# Personality Development and Behavior in Adolescence: Characteristics and Dimensions

Valentina Milenkova [1,2,*] and Albena Nakova [2]

1   Department of Sociology, South-West University "Neofit Rilski", 2700 Blagoevgrad, Bulgaria
2   Institute of Philosophy and Sociology, Bulgarian Academy of Sciences, 1000 Sofia, Bulgaria;
    albena_nakova.manolova@abv.bg
*   Correspondence: vmilenkova@gmail.com

**Abstract:** This article aims to present the specifics and characteristics of personality development during adolescence in light of the family's influence on the adolescent's self-esteem, self-perception, and behavior. Self-concept holds a particular importance in an individual's psychological and social development and expression. Self-concept contributes to an individual's communication, personal confidence, and independence. The objectives of the article are as follows: to trace some basic aspects of the influence of family on personality and its formation; to present concrete empirical dimensions of the Self-concept of students in a Bulgarian environment as well as their relationships with their parents; to show whether there are differences among the main ethnic groups in terms of their Self-concept and parental influence. The empirical analysis of the article is based on research conducted with secondary school students in 2018 within the framework of the "Modern Bulgarian Education: Status and Deficits" project, of which there were 130 Bulgarian, 70 Turk, and 70 Roma students aged 14–16 and of both sexes. The students were randomly selected from several Bulgarian secondary schools. To establish Self-images and the formation of various qualities, we used a method highlighting the types of personality tests used to register Self-concept profiles, including various personal qualities in different areas of personality expression. Students rate the qualities as real and as their desire to possess them on a five-degree scale from "1"—I do not possess at all to "5"—I completely possess the corresponding quality. The survey results showed that young people to a large extent tend to strive for the construction of their Self-image as open to sociability, contacts, and communication to attain affirmation among peers and autonomy. The article also analyzes assessments of parents' involvement in children's lives as well as the importance of family as a value. The main conclusion of the article is that upbringing in the family is key to the development of a child's Self-concept and success, the value structure and emotional state, as well as overall personal activity.

**Keywords:** parental influence; child; family upbringing; Self-concept; child development



## 1. Introduction

The topic of personal development and Self-concept is frequently discussed in the scientific literature. It is of particular relevance to society as childhood and adolescence are the periods in which personalities are formed. In the processes of personality development, family plays one of the main roles. A key characteristic of the family as a social community is that it implies close relationships, emotional and immediate connections, and providing support and assistance between members. The family is a complex and dynamic organism that underpins the social system and influences both its members and institutional relationships in local, regional, and national contexts. Families are characterized by some level of income that describes their economic status. On the other hand, they are distinguished by their ethnic and cultural specificity, including the systems of values and cultural models that follow. Families are also characterized by a certain pattern of relationships and contacts

that they create. They form an environment that provides understanding and support. Therefore, it could be said that families today are very different and their habitus—as a set of values, attitudes, and ideas—is what forms their identity. Family habitus reflects economic as well as cultural and psychological aspects, some of which are passed down through generations. For this reason, family has always had and still has great importance today. In the modern situation of changes and challenges, acute deficits of the culture of communication and attention to each other, as well as highly competitive environments and stress, people often feel insecurity. Therefore, family is of great importance in the context of globalization, as individuals of all ages need stability and confidence, which are built on family support and empathy. The role of the family is especially strong in childhood and adolescence when parents consciously and unconsciously influence their children.

The current article focuses on the influence of family on child development. Parents transmit models and values to their children and create an environment of support and understanding that is important at all stages of their life course. In this sense, the style of family upbringing is of primary importance. Positive relationships in the family form a positive attitude towards the world and oneself, and vice versa—they can form low self-esteem, high anxiety, and disorientation. The present article is aimed at researching family interactions and their projections in children's personal self-evaluation, attitudes, and Self-concept. Self-esteem is an important factor in personality development; it affects emotional well-being, and the ability to adapt and to communicate with adults and peers. The aims of the article are as follows: to trace some basic aspects of family influence on personality and its formation; to present specific empirical dimensions of the Self-concept of students in a Bulgarian environment, as well as their relationships with their family; and to show whether there are differences among the main ethnic groups in terms of their Self-concept and parental influence. The main research questions of the article are related to rethinking the influence of parents on the personal development of children, as well as looking for the effects of this influence in different dimensions of the Self-concept and personal evaluations in adolescence. The conducted research traces some socio-psychological characteristics of both the Self-concept and the existing family environment as a basis for adolescent development.

Our understanding is that upbringing in the family is key to the development of the child's self-awareness and success, their value structure and emotional state, as well as their overall personal activity.

*Theoretical Considerations*

The family creates a specific environment and conditions for cooperation and empathy [1], which are "a source of parental support" [2]. In the family, the child learns different attitudes, builds self-confidence and adapts styles of thinking that determine central orientations and statuses [3]. In this way, the models of relations with the environment and conditions are built and confirmed, which are decisive in personal terms.

The family is the first space for the upbringing of children's habits of social behavior and includes everyday care, behavioral patterns, and value standards. The experience of communication and relationships with parents and close adults determines the child's relationships with other people. Specific conditions of family upbringing can form attitudes and motives, hindering or favoring children's abilities, skills, and qualities. Achievement motivation is influenced by parenting styles related to behavioral control, emotional support, encouragement of success, or punishment for failure. If adults are attentive to nurturing actions, this leads to the establishment of stable attachment and effective forms of interaction [4].

One of the important studies on the influence of family on the child's development is Coopersmith's [3], in which he highlights various factors such as family size, order of birth, relationships with siblings, peers, features of the mother's and father's personalities and their relations with them, which are fundamental to the character and effectiveness of

family influence. Each of the listed factors affects the frequency of contact and the strength of the relationship between parents and children.

According to Coleman [2], parental influence and family ties are stronger in two-parent families with fewer children, where parents have greater ambitions and plans for their offspring. These conditions favor greater parental attention, more hours spent with children, and greater empathy. Child–parent relations are a prerequisite for the development of the child's personality [5]. An adequate level of relationships in the family helps the child to develop social experience and affects the formation of the emotional-will sphere. Communication with the child provides enrichment of the child's consciousness and determines its structure, development of various mental processes, consciousness, and self-awareness. Family provides the maximum duration of interpersonal relationships and knowledge of self and other family members [6].

The style of family communication, as well as the values of the family, are of great importance. As a socio-psychological concept, the style indicates a set of ways and forms of interaction. Two criteria are distinguished: the degree of emotional closeness, the warmth of the parents towards the child (love, acceptance, empathy, and understanding), and the degree of control over behavior—high, with many restrictions and prohibitions, or low, with minimal restrictions and prohibitions.

Factors are as follows: internal and external activities of the family, sharing of household duties, budget, relationship between spouses, and stress factors of different origins in home conditions that create additional emotional tension and anxiety.

The family environment is a space of active entry of the child into the social world and of the establishment of diverse relationships with adults and peers. It is important to provide the correct direction to the child's emotional development and to awaken humanity, a desire for cooperation, and positive self-affirmation [7].

Inability to interact with others is described as social insecurity manifested in non-communicativeness, autism, and hypoactivity. Hypoactivity and hyperactivity are forms of inadequate protective mechanisms of children's social insecurity. Therefore, one of the important tasks is the formation of confidence and a positive attitude towards oneself [3]. The formation of confidence depends on many conditions, both objective (relationships in the family, environment, and national and religious characteristics) and personal (temperament and nervous system).

The child perceives the love of those around him/her physically, intellectually, and emotionally [8]. In addition, when he/she does not receive this, defensive reactions and social fear develop in various forms with anxiety, worry, guilt and/or anger, which are associated with low self-esteem, expectation of failure, and increased dependence on others. Self-doubt is one of the personality traits that hinders the child [9], and emotional experience can have a positive or negative orientation.

Creating conditions for positive emotions in the child is very important on the part of adults. The child gradually begins to understand the surrounding world and realizes his/her place. This gives rise to new motives for behavior. On the other hand, feelings and emotions will develop, which ensure the effectiveness of these motives as well as the stability of behavior and its independence from the change in external circumstances. All sides of personality are in close unity and influence each other [6].

In a family, the child strives to be like adults and realizes not only the specific actions and qualities but also desires, experiences, and motives that unite and those that consolidate the individual. The child receives different influences from others: evaluations, remarks, approval, or disapproval. The whole mental life develops under the influence of the evaluation of others: new experiences, knowledge, and skills are evaluated by others. The child begins to look for evaluation of their actions and to expect support of their correctness [10].

The assessment of the adult plays a stimulating role, mobilizing the child's efforts towards obtaining a result. Low ratings of adults have negative consequences and reduce the independence and initiative of the child.

A child's self-esteem [11] is related to the degree of autonomy and support a child receives from parents. Self-esteem is the image and assessment of oneself and one's qualities, capabilities, and relationships with others; it is the most important component of personality. Self-assessment is the basis for the formation of critical thinking—the ability to analyze reality. It refers to awareness of the person and of one's physical and mental abilities, motives, goals, and actions [12]. It indicates the extent to which a person believes that he or she is capable, significant, successful, and valuable [13].

The formation of self-esteem begins at the age of 3–4, when the child begins to operate with the concepts of "good–bad", "right–wrong", "beautiful–ugly". The basis of self-esteem is laid by parents. Self-esteem is the result of constantly comparing what the individual observes in him/herself with what he/she sees in other people and with assumptions about how they assess him/her.

According to Coopersmith [3], self-esteem is a "positive and negative attitude towards oneself". Moreover, this means approval or disapproval of oneself and shows confidence in one's own capabilities and qualities. In the self-assessment test, Coopersmith includes 58 assessment indicators for children and 50 for adults [14]. According to Coopersmith, there are several important conditions that underlie positive self-esteem: (1) Unconditional acceptance of children by adults, creating an environment of love and warmth. (2) Create a framework with clearly defined boundaries and requirements that are fair, pressure-free, and negotiable. (3) Respectful and serious attitude of adults towards the personality of the child. (4) Parents are role models—they are living examples of efficacy and self-respect.

Self-esteem plays an important role in self-regulation mechanisms, determines the level of attachment, and influences the development of interpersonal relationships [15]. Self-esteem is the core of the process of self-knowledge, an indicator of the individual level of development, and integrates the personal aspect that is organically included in the development process.

Closely related to self-esteem is the Self-concept of personality, which has several different manifestations related to understanding the present and openness to change. The process of Self-concept development, regardless of the existing general regularities, has its individual characteristics in each child. Self-concept is characterized by inner knowledge, which is manifested in the understanding and evaluation of one's actions, deeds, thoughts, and experiences with the help of language, and their attitude towards nature and other people.

According to Burns [16], Self-concept is a person's awareness of his/her actions, thoughts, feelings, interests, and position in society, i.e., awareness of oneself as a person placed in a certain environment, which has its own specifics in different age stages. It is a complex process mediated by self-knowledge, unfolding over time, and related to the movement from single situations through the integration of similar images to the creation of one's own self as a subject different from other people. The multi-level process of self-knowledge is connected with the diverse experiences that are refracted through the emotional-value attitude of the person towards oneself.

Thus, Self-concept includes Self-image on the one hand and self-evaluation on the other, which is based on several components: internalized social evaluations, comparing the Self-image with the desired Self-image, and individual self-evaluation of actions and results [17]. This means the formation of qualities that do not only follow external requirements but consist in a specific internal autonomy of the person, objectified in independent decision-making, directed thinking, and making choices that are not arbitrary.

Therefore, in the behavior of adolescents, a balance must be maintained between the stimulation of manifestations related to an adaptive strategy of expression aimed at building qualities such as "discipline" and "compliance with the requirements of adults", and manifestations showing an active strategy oriented towards qualities such as "criticality", "initiative", "self-confidence", and "striving for self-improvement". According to some authors, Self-concept is very important; it may even be the most important variable for predicting academic success and, according to others, is important not only for academic

success but for overall behavior [18]. Self-concept is one of the important results regarding the influence of family; in addition, its different dimensions show different personality characteristics that refer to spheres of personality expression.

In the context of the thesis about the influence of family on personal development, a study was conducted with school-aged youth aimed at registering various aspects of Self-concept as well as the relations of adolescents with the family environment.

## 2. Material and Methods

In the conducted research, a combination of two main scientific approaches is applied—the approaches of sociology and social psychology. The social psychological approach includes the study of Self-image and its two dimensions of Self-real and Self-ideal. The sociological approach is aimed at revealing the role of parents in the formation of the Self by tracing the social activities of family participation and involvement in the lives of adolescents.

To establish ideas of the Self and the formation of various qualities, the socio-psychological method developed by Ivan Paspalanov of the type of personality tests was used to register the profiles of Self-concept [19] (pp. 60–74), which includes the elaboration of 17 personality qualities in different areas of personality expression (Table 1).

**Table 1.** Personal qualities and areas of expression.

| Areas of Expression | Qualities |
| --- | --- |
| Work | Diligence |
|  | Organized |
| Social control | Discipline |
|  | Respectful of elders |
| Learning | Inquisitiveness |
| Creativity | Original thinking |
| Autonomy | Responsibility |
|  | Independence |
| Self-attitude | Confidence |
|  | Self-criticism |
| Attitude towards success | Persistence |
| General life attitude | Sense of humor |
|  | Optimism |
| Ethics in communication | Politeness |
|  | Honesty |
| Social activity | Criticality |
|  | Sociability |

Source: [19] (pp. 60–74).

The sociological method includes surveys among students from different schools in the country. The questions constructed in the questionnaires are the authors' own contributions.

Both methods were applied to students from secondary schools collected randomly from several cities: 170 students from Sofia (79th Secondary School, 40th Secondary School, and 56th Secondary School) and 100 students from the country (50 from Blagoevgrad—7th Secondary School; 50 from Targovishte—1st Secondary School and 2nd Secondary School).

A special focus of the analysis was the self-assessment of different ethnic groups, as ethnicity is brought up as one of the lines differentiating relationships within the family. Ethnicity also affects people's education, employment, and economic status. Thus, ethnicity was important for us to look for differences in Self-concept profiles as well as in the environment that the family provided to children. Therefore, we focused on several cities in the country where there is a significant representation of the three main ethnic communities—Bulgarians, Turks, and Roma. Thus, we focused on Blagoevgrad (with a significant representation of Roma ethnicity), Targovishte (with a significant representation of both ethnicities—Roma and Turks), and Sofia (with a representation of Bulgarians and Roma). The number of pupils surveyed (the sample) was calculated as the percentage of

pupils from each of the three ethnic groups in each of the schools surveyed compared to the percentage of children aged 14–16 in the overall population structure in each of the cities surveyed according to the 2011 census data in the Republic of Bulgaria. This generated a sample of 270 students aged 14–16 of both genders with an emphasis on ethnicity (Table 2).

**Table 2.** Description of the participants.

| Social Characteristics | Numbers | Numbers | Numbers |
|---|---|---|---|
| Ethnicity | 130 Bulgarians | 70 Turks | 70 Roma |
| Gender | 140 girls | 130 Boys | |
| Age | 30: 14 years | 100: 15 years | 140: 16 years |
| Residence | 170: from Sofia | 100—from the country | |

In order to conduct the survey, permission was requested from the relevant Regional Inspectorates of Education in the three cities as well as permission from school principals and parents' consent. The selection of the students was made based on their willingness to participate in the survey. The researcher entered the classroom of the respective school and presented the objectives of the study. Students who were willing to participate (following the calculated sample) completed the questionnaire. The entire study was carried out in accordance with ethical norms, and consent was obtained from the Ethical Commission of the SWU. The questionnaire (Appendix A) included questions related to Self-concept as well as questions related to the role of parents in the upbringing of children and their personal development. At the beginning of each fieldwork with the survey, a team member explained the purpose of the study and provided further clarification on each of the questions. The students were made aware that the information was anonymous and used for scientific purposes only. Respondents were informed that their participation was voluntary. At the beginning of the questionnaire (Appendix A), an address was made to the researched person, where the objectives of the study and the nature of the information obtained were also presented.

The qualities included in the personality test (first part of the research) were accompanied by instructions corresponding to the 2 dimensions of Self-image: Self-real and Self-ideal. Respondents assessed each quality in the different series as possessed or desired on a five-level scale (e.g., To what extent do I possess (or desire to possess) the given quality?: 1—"Very little", 2—"Relatively little", 3—"Medium", 4—"Relatively much", 5—"Very much"). The instructions were written in an understandable form and presented as the individual's self-description of "How I know myself".

The tests for investigating the Self-concept in its two dimensions of Self-real and Self-ideal have the following form:

I have the following qualities:

| | | | | | |
|---|---|---|---|---|---|
| Persistence | 1 | 2 | 3 | 4 | 5 |
| Honesty | 1 | 2 | 3 | 4 | 5 |
| Responsibility | 1 | 2 | 3 | 4 | 5 |
| Senseofhumor | 1 | 2 | 3 | 4 | 5 |
| Self-criticism | 1 | 2 | 3 | 4 | 5 |
| Diligence | 1 | 2 | 3 | 4 | 5 |
| Discipline | 1 | 2 | 3 | 4 | 5 |
| Sociability | 1 | 2 | 3 | 4 | 5 |
| Confidence | 1 | 2 | 3 | 4 | 5 |
| Respectfulofelders | 1 | 2 | 3 | 4 | 5 |
| Organized | 1 | 2 | 3 | 4 | 5 |
| Independence | 1 | 2 | 3 | 4 | 5 |
| Inquisitiveness | 1 | 2 | 3 | 4 | 5 |
| Originalthinking | 1 | 2 | 3 | 4 | 5 |

| | | | | | |
|---|---|---|---|---|---|
| Optimism | 1 | 2 | 3 | 4 | 5 |
| Politeness | 1 | 2 | 3 | 4 | 5 |
| Criticality | 1 | 2 | 3 | 4 | 5 |
| I would like to have the following qualities: | | | | | |
| Persistence | 1 | 2 | 3 | 4 | 5 |
| Honesty | 1 | 2 | 3 | 4 | 5 |
| Responsibility | 1 | 2 | 3 | 4 | 5 |
| Senseofhumor | 1 | 2 | 3 | 4 | 5 |
| Self-criticism | 1 | 2 | 3 | 4 | 5 |
| Diligence | 1 | 2 | 3 | 4 | 5 |
| Discipline | 1 | 2 | 3 | 4 | 5 |
| Sociability | 1 | 2 | 3 | 4 | 5 |
| Confidence | 1 | 2 | 3 | 4 | 5 |
| Respectfulofelders | 1 | 2 | 3 | 4 | 5 |
| Organized | 1 | 2 | 3 | 4 | 5 |
| Independence | 1 | 2 | 3 | 4 | 5 |
| Inquisitiveness | 1 | 2 | 3 | 4 | 5 |
| Originalthinking | 1 | 2 | 3 | 4 | 5 |
| Optimism | 1 | 2 | 3 | 4 | 5 |
| Politeness | 1 | 2 | 3 | 4 | 5 |
| Criticality | 1 | 2 | 3 | 4 | 5 |

The average score that each quality receives was calculated. Depending on the average score obtained, rank orders were constructed for the two dimensions of Self-image. For the purpose of the analysis, the qualities ranked highest and lowest were separated using the mean and standard deviation of the scores in each row [20]. Therefore, for each dimension of Self-image, different qualities fell in the different subgroups of individuals that occupied the highest and lowest ranks.

In the second stage of processing, a Kendall's rank correlation coefficient was calculated between the rank series of the two Self-image dimensions in each group. This allowed us to make a quantitative assessment of the correlation between the ranking of qualities in the two dimensions of the Self-image.

Respondents passed through two stages of self-assessment corresponding to the dimensions of Self-real and Self-ideal. The average score that each quality received was calculated and ranks were compiled. Through the ability to self-describe, a generalized picture of the real and ideal image (that respondents aspire to) was obtained, as different qualities correspond to different areas of expression; reflect emotional status; and follow social relationships, goals, and adaptation. By assessing the various personality traits, Self-image was measured, as well as one's ability to control one's inner world and levels of interaction with others.

In examining self-esteem, as well as other aspects of parental influence in personality formation, ethnic profiles were also studied to show whether there are significant differences in parental influence by ethnicity.

## 3. Results

### 3.1. The Self-Concept of Personality

We track which qualities were given ranks from 1 to 5 by the students in the entire studied population in terms of their real Self and ideal Self (Table 3).

An important aspect of the Self-concept is the achievement of correspondence between the qualities that the individual actually possesses and those to which he/she strives for in the plan of the ideal Self, because in the comparison between the real and the ideal Self, there is no complete coverage. To obtain a clearer picture of the relationship between the real and the ideal Self, we need to look at the rank correlation coefficients between the Self-image profiles. From a socio-psychological point of view, there must be some overlap between the real and the ideal Self in order for the ideal self to have a motivating function and stimulate appropriate behavior. With a large overlap, this motivating function cannot

be fully accomplished. When the coefficient is very low, it means that there is no correlation in the ranking of the qualities in the segments of the Self-image [20] (pp. 78–95). The rank correlation coefficient indicates the similarity of the ordering of the data when ranked within the variables. The rank correlation coefficient (Kendall) $\tau$ between the Self-real and Self-ideal aspects for the entire sample is $\tau = 0.60$, $p = 0.01$. This denotes the presence of a significant correlation between the indicated sections of the Self-image. The correlation coefficients are specified for the three ethnic groups (Table 4).

**Table 3.** Rankings of Self-real and Self-ideal qualities of students.

| Quality | Self-Real | Self-Ideal |
|---|---|---|
| Persistence | 1 | 6 |
| Honesty | 2 | 2 |
| Responsibility | 3 | 1 |
| Sense of humor | 4 | 7 |
| Self-criticism | 5 | 3 |
| Diligence | 6 | 8 |
| Discipline | 7 | 9 |
| Sociability | 8 | 4 |
| Confidence | 9 | 10 |
| Respectful of elders | 10 | 11 |
| Organized | 11 | 5 |
| Independence | 12 | 13 |
| Inquisitiveness | 13 | 14 |
| Original thinking | 14 | 12 |
| Optimism | 15 | 15 |
| Politeness | 16 | 16 |
| Criticality | 17 | 17 |

$\tau$ (270) = 0.60, $p = 0.01$.

**Table 4.** Rank correlation coefficients between Self-real and Self-ideal by ethnicity.

| Ethnicity | Coefficient ($\tau$) | $p$ |
|---|---|---|
| Bulgarians | 0.59 | 0.02 |
| Roma | 0.61 | 0.01 |
| Turks | 0.60 | 0.01 |

The correlation coefficients for students from different ethnic communities are similar, which indicates similarity between the real and desired qualities of the self; the qualities are connected and there is continuity in the self-definition of the respondents.

We will follow how the ranking of the qualities develops in the two sections of Self-image according to ethnicity.

In the group of Bulgarians (Table 5), the following similarity is noticeable: the qualities that have ranks from 1 to 5 in Self-real remain in Self-ideal while only changing their ranks (the exception is "persistence", which remains in first place).

**Table 5.** Rank correlation between Self-real and Self-ideal—Bulgarian students: the first five qualities.

| Self-Real | Rank | Self-Ideal | Rank |
|---|---|---|---|
| Persistence | 1 | Persistence | 1 |
| Honesty | 2 | Self-criticism | 2 |
| Responsibility | 3 | Honesty | 3 |
| Sense of humor | 4 | Responsibility | 4 |
| Self-criticism | 5 | Sense of humor | 5 |

$\tau$ (130) = 0.59, $p = 0.02$.

This refers to a distinct orientation of the Self toward assertion and autonomy.

It is striking that four of the five qualities listed with the highest ranks among students of Roma origin (Table 6) are also found among Bulgarians: "sense of humor", "persistence", "honesty", and "responsibility".

**Table 6.** Rank correlation between Self-real and Self-ideal—Roma students.

| Self-Real | Rank | Self-Ideal | Rank |
|---|---|---|---|
| Persistence | 1 | Responsibility | 1 |
| Diligence | 2 | Honesty | 2 |
| Sense of humor | 3 | Persistence | 3 |
| Honesty | 4 | Sociability | 4 |
| Responsibility | 5 | Persistence | 5 |

$\tau$ (70) = 0.61, *p* = 0.01.

Two new qualities, "sociability" (Self-ideal) and "diligence" (Self-real), appear. "Diligence" (2) in Self-real receives a rank of 8 in Self-ideal. The main keywords describing the set of real qualities of Roma children are "ethics in communication" and "autonomy", and for ideal qualities are "ethics in communication", "autonomy", and "sociability".

In the children of Turkish origin (Table 7), in Self-real (ranked from 1 to 5), there are four qualities that are also found in the other two ethnic groups: "persistence", "honesty", "responsibility", and "sense of humor".

**Table 7.** Rank correlation between Self-real and Self-ideal—Turkish students.

| Self-Real | Rank | Self-Ideal | Rank |
|---|---|---|---|
| Persistence | 1 | Diligence | 1 |
| Honesty | 2 | Organized | 2 |
| Sense of humor | 3 | Honesty | 3 |
| Responsibility | 4 | Sense of humor | 4 |
| Diligence | 5 | Sociability | 5 |

$\tau$ (70) = 0.60, *p* = 0.01.

The qualities "diligence" and "organized" are among the five qualities with the highest ranks in the Self-ideal. "Persistence" from position 1 in Self-real moves to position 10 in Self-ideal.

*3.2. Family and the Life World of Adolescents*

In order to trace the influence of parents on the formation of the Self-concept of adolescents, we asked the following question: "To what extent have your parents contributed to the formation of you as a person and to the development of your various qualities and skills?". Respondents rated their parents' influence on a scale from 1 to 5, where 1 is the lowest and 5 is the highest influence. The following distributions were obtained:

- A total 83.3% of the persons gave a score of "5", they estimate that their parents have the greatest influence on their personal development and qualities. By ethnicity, this percentage is distributed as follows: 93.8% (Bulgarians), 81.4% (Turks), and 65.7% (Roma).
- In total, 7.4% of the adolescents gave a score of "4", which means that their parents have a great influence on the development of their personal qualities.
- A total 5.6% of the students rated "3", which refers to "moderate" parental influence.
- Only 3.7% gave a score of "2", meaning "weak" parental influence.
- A score of "1", indicating a lack of parental influence, was not indicated by the adolescents.

The survey included a set of questions relating to the importance of family in the everyday world of adolescents in order to track their assessments of parental involvement in their children's lives, the dimensions of this involvement, as well as the significance of family as a value. In this way, the influence of parents is tracked in specific daily activities that show how effective the parent–child relationship is through the evaluation of the teenagers themselves. On the other hand, in the sample, the presence of young people from the three ethnic communities allows us to compare the influence of family as a factor of personal development in an ethnic context.

To the question "Do you think family is important to you?", the following responses were received (Table 8).

**Table 8.** Do you think family is important to you?

| Answer | Total (%) | Bulgarians (%) | Turks (%) | Roma (%) |
|---|---|---|---|---|
| Yes | 81.5 | 86.2 | 85.7 | 68.6 |
| To some extent | 14.8 | 12.3 | 11.4 | 22.9 |
| No | 3.7 | 1.5 | 2.9 | 8.6 |

We expanded on the topic of the role of family in creating a living environment with the following question: "Do you celebrate different holidays with your parents and relatives: birthdays, family holidays, public holidays?" (Table 9).

**Table 9.** Spending different holidays with family.

| Holidays | Total (%) | Bulgarians (%) | Turks (%) | Roma (%) |
|---|---|---|---|---|
| Birthdays | 85.2 | 96.2 | 92.9 | 57.1 |
| Family holidays | 83.3 | 92.3 | 85.7 | 64.3 |
| Public holidays | 92.6 | 94.6 | 85.7 | 95.7 |

It is noteworthy that in Bulgarian families there is a high percentage of celebrating various events, including family events and public holidays; almost all adolescents from the Bulgarian ethnic group included in the sample stated that in their families, celebrating holidays fulfills an integrating function.

Special attention is required by the high percentage of Roma youth (95.7%) who indicated public holidays as an occasion to gather relatives. In fact, many of the public holidays, such as St. George's Day and Basil's Day—the Roma New Year, have become a symbol of Roma ethnicity itself and have great significance for both the community and the family. In general, the Roma family is identified with the extended family, including all relatives, which imposes the unifying function of the family and is understood as a kinship community.

In addition to the celebration of various holidays in the family, the influence of parents is also manifested in various other activities through which care, support, models of behavior and understanding, etc., are carried out. We asked the following question: "What common activities do you do in your family: watching TV, discussing various topics; family games; household activities?" (Table 10).

**Table 10.** Conducting activities with family.

| Activities | Total (%) | Bulgarians (%) | Turks (%) | Roma (%) |
|---|---|---|---|---|
| Watching TV | 81.5 | 95.4 | 71.4 | 65.7 |
| Discussing various topics | 66.7 | 96.9 | 42.9 | 34.3 |
| Family games | 70.4 | 96.2 | 42.9 | 50.0 |
| Household activities | 63.0 | 40.0 | 92.9 | 75.7 |

Regarding family activities, the following could be said:

- The activity most common for the whole sample was "watching TV", indicated by 81.5% of individuals.
- Within the Bulgarian ethnic group, almost equally with large accumulations are the following activities in the family: "watching TV" (95.4%), "discussing different topics" (96.9%), and "family games" (96.2%).
- In the remaining two ethnic groups, the largest accumulations are in "doing household activities": for the Turkish community—92.9% and for the Roma community—75.7%. This refers to various activities related to agriculture, which are part of the household, widespread among Turks, or "garbage collection", typical of Roma communities. Caring for younger siblings as well as elderly relatives is also part of the family responsibilities of children in Roma and Turkish families due to their large families.

Within the package of questions that present the importance of family in the life world of adolescents, the degree to which they participate in the family environment and communicate with parents and relatives, as well as the level of integration in the family community, we asked one personal question: "Is there someone in your family with whom you can discuss personal matters?".

The share of positive answers for the entire sample is relatively high at 82.6%; the largest accumulations are among students from the Bulgarian ethnic community—92.3%; among the Turks—78.6%; and for Roma youth—68.6%.

In addition to the examined questions, we asked students to rank five different values according to their significance: family, education, work (profession), money, and friends, which represent various life phenomena. The purpose of the values ranking was to track the attitude towards personally important things and their assessment and importance for adolescents.

The following rankings by ethnicity were obtained (Table 11).

**Table 11.** Ranking of values.

| **Bulgarians** | **Turks** | **Roma** |
|---|---|---|
| Family (1.5) | Family (1) | Money (1.5) |
| Friends (1.5) | Education (2) | Family (1.5) |
| Money (3) | Work (profession) (3) | Friends (3) |
| Education (4) | Friends (4) | Education (4) |
| Work (profession) (5) | Money (5) | Work (profession) (5) |

It is noticed that family in all three ethnic groups is placed in first place, which means it has a great value and is defined as a personally significant factor for the adolescents.

## 4. Discussion

This paper has confirmed the concepts of Self-image discussed in the theoretical framework (Coopersmith, Aloia, Erikson, Burns, Bornstein) as central to the system of personality factors that influence the individual's potential. The formation of Self-image plays a key role in the general structure of personality [12].

The ability to interact and create social relationships is important because it enables students to develop their communicative skills, to be active, and to build an integrated personality.

The family is likewise of great importance for their formation. What place do the personal qualities proposed for assessment occupy in the general concept of family influence? Indisputably, each of the qualities in question is to a lesser or greater degree subjected to the influence of parents and the family environment. We singled out a group of qualities that indicate the adolescent's striving for autonomy. These are "sociability", "independence", "responsibility", "optimism", "self-criticism", "sense of humor", and "original thinking". These characteristics indicate a desire for autonomy, sociability, ethics in communication,

and attitude toward oneself. They are very important on a personal level and emphasize the desire for self-reliance and independence, which are critically significant for adolescents but also indicate the degree of maturity of the family, whose role is to provide some main elements of upbringing for their children; this educational influence includes building motivation for self-expression and independence. Our understanding is that the desire for autonomy, independence, and self-reliance, among others, is inspired by family and indicates the style of upbringing that parents give their children. We ranked the domains of expression and their corresponding qualities in order to then track the respondents' ratings in the sections on Self-real and Self-ideal.

In the context of Self-real, the analysis of the qualities that respondents rated highly revealed a desire for ethics in communication and for autonomy, and defined the respondents' general attitude to life and attitude to success. The set of qualities can be seen as related to personal success, motivation, and autonomy. The three most highly valued qualities are as follows: "persistence" (1)—related to "attitude to success"; "honesty" (2)—related to "ethics in communication"; and "responsibility" (3)—related to personal "autonomy". In the middle range of the ranking, the qualities "respectful of elders" (10), "independence" (12), and "optimism" (15) remain in the lower range. In the domain of Self-real, "sociability" (8) and "confidence" (9) are qualities directly related to social activity and attitude to oneself. With regard to the respondents' self-descriptions, these are among the ten highest-rated qualities, which suggests a certain attitude to individual activity.

Certain changes are observed in the domain of Self-ideal. First, two new qualities appear (in the range from 1 to 5)—these are "organized" and "sociability"; by comparison, "persistence" and "sense of humor" are ranked lower. We see that success attitudes and ethics in communication give way to individual activity. This suggests that self-expression, self-affirmation, and social activity are much more important for adolescents. The main conclusion is that the dominant striving during adolescence is for psycho-social identity, self-affirmation among peers, and recognition from peers [7].

The general overview of the qualities appearing in the different sections related to Self-image leads to the following conclusions:

- Two qualities that are ranked lower in the field of Self-real move to higher positions in the field of Self-ideal: "organized" (from 11 to 5) and "sociability" (from 8 to 4). These qualities show a desire for personal development and self-affirmation among peers.
- The ranks of another group of qualities move downward in the area of Self-ideal: "persistence" (from rank 1 to rank 6) and "sense of humor" (from 4 to 7).
- For a third set of qualities, no significant changes are observed across the areas of Self-real and Self-ideal. Their rankings either do not change at all—"self-criticism" (17), "optimism" (15), "politeness" (16), and "honesty" (2), or change insignificantly (by one or two positions)—"original thinking" (from 14 to 12), "inquisitiveness" (from 13 to 14), "independence" (from 12 to 13), "confidence" (from 9 to 10), "discipline" (from 7 to 9), "diligence" (from 6 to 8), and "respectful of elders" (from 10 to 11).

The obtained results indicate that personal qualities and the development of the Self-concept among adolescents are most strongly influenced by age and less influenced by ethnicity. Adolescents are apparently more concerned about their relationships with peers, as these relationships place them in a new role within their social environment.

The results of the personality test showed a pronounced striving for qualities related to ethics in communication, sociability, and autonomy, i.e., to domains related to interaction with peers and decreasing control [21] (pp. 248–258). The search for autonomy supports positive self-esteem and is related to personal self-expression and self-affirmation. All of this reflects the attained level of self-confidence and the ability to cope with emotional vulnerability [22]. Alongside the search for independence, the importance of interpersonal comparisons and relationships with peers increases [23]. Informal relationships become a source of emotional support for adolescents experiencing common problems, such as the discovery of their personal significance, the search for a new social status, the emancipation from parental authority, and gender identification [24]. The ranking of the individual's

self-evaluations reflects the processes of external and internal differentiation as well as the hierarchical structure of the self in search of identity. Adolescents strive for qualities that allow them to achieve self-regulation, communication skills, and the ability to cope with their social world [11].

Our knowledge about the content and structures of the Self-image enables us to also trace the socio-educational influences at play in the construction of the Self-concept.

In general, it may be said that family is assessed as a value by both the sample as a whole and by ethnic groups within the sample. The results clearly show the effect of family upbringing and the role of the parents as factors in the development of personal qualities. Thus, adolescents assess how important their parents were and continue to be for them; this gives us reason to consider the Self-concept of adolescents as a result, among others, of parental influence.

The survey results suggest that family is important both within the entire sample and for ethnic groups; the largest accumulations of this opinion are registered among Bulgarians and Turks. The family is considered important not only for its perceived influence on the respondents' personal development but also as a value passed down through the generations. We find that family represents an integrating reality and the majority of families value the experience of this integration, which the adolescents definitely feel to be important. Thus, along with the emphasis on autonomy and personal independence that we found in the Self-concept, we also observe high assessment of the importance of family. This assessment suggests active parental participation in the lives of their children and the search for common points of contact. The surveyed adolescents definitely believe the parents have made a significant contribution to the personal development of their children.

Based on the results obtained through both types of methods, we may propose a topic of discussion related to adolescence as a specific age group. On the one hand, autonomy and personal independence have not yet led to emancipation of the adolescent's personality. On the other hand, family and parents are found to be influential and figures of authority in terms of the upbringing they give their children.

It can be said that in the families of Bulgarian ethnicity, a probably larger number of adolescents feel their parents are providing personal support and can share with them. This is due to this ethnic group's greater openness to modern values, which predisposes young people to communicate with their parents and discuss various topics with them. Thus, along with family's importance with regard to the qualities of autonomy and personal independence, which we identified in the analysis of the Self-concept, the integrating power of family is also emphasized. This integrating force is evident in the celebration of various events, the performance of common activities (differentiated according to ethnic specificity), and in the possibility of discussing various topics. This can also be interpreted as an indicator of the acceptance of parental influence in the specific life situation of early adolescence. Family provides stability and inspires a feeling of security as the basis of bonding capital, which is a form of social capital related to ties between family members [25]. The fact that family is mentioned as a value by all three ethnic groups (and that the survey results indicate the role of parents in the development of their children) can be seen as a re-evaluation of the role of family and its importance. In the context of globalization and social crises, the importance of family is growing and being reconsidered by young people.

## 5. Conclusions

The present article focused on presenting different dimensions of Self-concept at school age as well as tracing the role of parents in building social habits and value patterns. The article confirmed the concepts considered in the theoretical part about the influence of family as an environment for cooperation and empathy. The theses suggesting that family is a space for the formation of behavioral patterns and value standards were confirmed. The article has shown that the traced aspects of communication in the family; of the joint activities that are carried out—including the line of holidays, which become a unifying element of the family environment; as well as the examination of the socio-psychological

characteristics of the Self-concept, reveal the presence of an active family environment and nurturing parental influences.

The results show that when ranking the qualities according to ethnicity, there is a constancy of three qualities regarding attitude to success, ethics in communication, and personal autonomy—"persistence", "honesty", and "responsibility", which are manifested in all three ethnic communities. In general, adolescents (especially from the Bulgarian ethnicity) emphasize qualities related to success and general life attitude. This means that young people to a large extent tend to strive for the construction of their Self-image as open to sociability, contacts, and communication to achieve affirmation among peers and autonomy. Other things should be taken into account here, such as the desire of young people to be independent and to resist control, which often creates tensions in communication with adults. The desire to form one's own opinion and assessment, different from those of teachers and parents, sometimes becomes an impulse often pursued for its own sake. We must realize, however, that young people find their self and arrive at important discoveries and explanations for themselves and for the world [7]. By developing confident, active, and stable young people, the social system becomes more resilient and its formative impacts are more effective [26].

In addition, the results of the study showed that families in Bulgarian conditions have positive influences on their children, from the point of view of creating unifying activities such as watching TV, discussing various topics, and participating in household activities. Holidays—family and official—are another important point of family bonding. It can be said that family and parents are influential; through their behavior and various integrating actions, they support the children's awareness and self-affirmation, thereby supporting their Self-concept. Adolescents' assessment that parents and family are important to them is an indicator of parental involvement in their children's lives and finding bonding and compatibility, which means that family is perceived as a source of support and empathy.

In this paper, some research limitations can be outlined related to the fact that the sample of the study included students from only three Bulgarian cities. Therefore, in our future research work, we envisage conducting a representative nationwide survey based on the same methodology to track the current state of adolescents' Self-concept as well as the influence of parents on personal development and behavior. Another research limitation was that the Self-concept test included only two qualities: responsibility and independence, expressing the autonomy of the individual. The latter, according to the results of the study, is of great psychological importance for adolescents. From this point of view, we plan to add more personality qualities representing autonomy—the idea of which arose as a result of the conducted research—in order to study it in more depth. This article confirms the role of family and parental influence as factors in adolescents' environments. This has significance both socially, in order to show the place of family as a factor in personal development, and scientifically, as continuity between authors working in the thematic field and interaction between the sciences studying it, and particularly between social psychology and sociology.

**Author Contributions:** Conceptualization, V.M. and A.N.; methodology, V.M. and A.N.; validation, V.M. and A.N.; formal analysis, V.M. and A.N.; investigation, V.M. and A.N.; resources, V.M. and A.N.; data curation, V.M. and A.N.; writing—original draft preparation, V.M. and A.N.; writing—review and editing, V.M. and A.N.; visualization, V.M. and A.N. All authors have read and agreed to the published version of the manuscript.

**Funding:** This research received no external funding.

**Institutional Review Board Statement:** The study was conducted in accordance with the Declaration of Helsinki and approved by the Ethics Committee of SWU, 17 March 2020.

**Informed Consent Statement:** Informed consent was obtained from all respondents involved in the study. By agreeing to participate in the survey, individuals provided their consent. As the results represent aggregated and anonymized data, no personal data about respondents and their personal opinions are disclosed.

**Data Availability Statement:** The results of this research have not been published elsewhere. The following supporting information can be downloaded at https://www.nsi.bg/census2011/ (accessed on 29 April 2023), Население (nsi.bg)—2011 Census data for persons aged 14–16.

**Conflicts of Interest:** The authors declare no conflict of interest.

## Appendix A

**Survey Questionnaire**

Dear respondent,

We are a team of researchers from various scientific institutions and we are conducting a sociological survey on the topic: "Modern Bulgarian education: state and deficits". We are turning to you with a request to participate in the research, which aims to study different aspects of the education and students' activities in modern Bulgarian society. The questions included in the questionnaire refer to your everyday life at school and your relationships in the family and with your peers. You have been included in our sample and your opinion is very valuable to us. The survey is anonymous and the information obtained is for scientific purposes only. Filling out the questionnaire will take you about 10–15 min.

Thank you in advance!

1.  **Do you like going to school? (Indicate one answer)**

    (a)    Yes
    (b)    Sometimes
    (c)    No

2.  **Do you understand the study material being taught? (Indicate one answer)**

    (a)    Yes, totally
    (b)    Yes, but I have a little difficulty in some subjects
    (c)    Yes, but I have serious difficulties in most subjects
    (d)    I do not understand the study material
    (e)    Other (Please write)

3.  **When you have learning difficulties, what do you do? (Indicate as many answers as you want)**

    (a)    I ask the teacher to explain
    (b)    I ask a classmate
    (c)    I rely on my parents to help me
    (d)    I go to private lessons
    (e)    I don't do anything

4.  **Do you have favorite subjects?**

    (a)    Yes, (Please write)
    (b)    No

5.  **Do you go to school regularly? (Indicate one answer)**

    (a)    Yes
    (b)    Sometimes
    (c)    No

6.  **For what reasons are you most often absent from school? (Indicate as many answers as you want)**

    (a)    Healthy reasons
    (b)    Family reasons
    (c)    I don't like going to school
    (d)    Economically reasons
    (e)    Something else (please write)

7. **Do your parents help you prepare for school? (Indicate one answer)**

   (a)  Yes, very often
   (b)  Yes, occasionally
   (c)  No

8. **What do you do in your spare time? (Indicate as many answers as you want)**

   (a)  I do sports
   (b)  I play computer games
   (c)  Meet with friends
   (d)  I help my parents
   (e)  Something else (please write)

9. **To what extent have your parents contributed to the formation of yourself as a person and to the development of your various qualities and skills? (Indicate one answer)**

   (a)  To a very large extent      1
   (b)  To a great extent            2
   (c)  On average                   3
   (d)  To a small extent            4
   (e)  They do not contribute       5

10. **Do you think family is important to you? (Indicate one answer)**

    (a)  Yes
    (b)  To some extend
    (c)  No

11. **Do you celebrate different holidays with your parents and relatives: birthdays, family holidays, public official holidays? (Indicate as many answers as you want)**

    (a)  Birthdays
    (b)  Family holidays
    (c)  Public holidays
    (d)  Something else (please write)

12. **What common activities do you do in your family? (Indicate as many answers as you want)**

    (a)  Watching TV
    (b)  Discussing various topics
    (c)  Family games
    (d)  Household activities
    (e)  Something else (please write)

13. **Is there someone in your family with whom you can discuss personal matters? (Indicate one answer)**

    (a)  Yes
    (b)  No

14. **Rank the following five different values according to their significance to you.**

    (a)  Family
    (b)  Education
    (c)  Work (profession)
    (d)  Money
    (e)  Friends

15. The next group of questions is about what qualities you possess. Rate each quality on a scale of 1 to 5, as "1" is the lowest degree of possession of the quality and "5" is the highest degree of possession of the quality.

    **I have the following qualities:**

    | | | | | | |
    |---|---|---|---|---|---|
    | Persistence | 1 | 2 | 3 | 4 | 5 |
    | Honesty | 1 | 2 | 3 | 4 | 5 |
    | Responsibility | 1 | 2 | 3 | 4 | 5 |
    | Sense of humor | 1 | 2 | 3 | 4 | 5 |
    | Self-criticism | 1 | 2 | 3 | 4 | 5 |
    | Diligence | 1 | 2 | 3 | 4 | 5 |
    | Discipline | 1 | 2 | 3 | 4 | 5 |
    | Sociability | 1 | 2 | 3 | 4 | 5 |
    | Confidence | 1 | 2 | 3 | 4 | 5 |
    | Respectful of elders | 1 | 2 | 3 | 4 | 5 |
    | Organized | 1 | 2 | 3 | 4 | 5 |
    | Independence | 1 | 2 | 3 | 4 | 5 |
    | Inquisitiveness | 1 | 2 | 3 | 4 | 5 |
    | Original thinking | 1 | 2 | 3 | 4 | 5 |
    | Optimism | 1 | 2 | 3 | 4 | 5 |
    | Politeness | 1 | 2 | 3 | 4 | 5 |
    | Criticality | 1 | 2 | 3 | 4 | 5 |

16. Now, please indicate what qualities you would like to possess. Each quality is rated from 1 to 5, as "1" being the lowest degree of desire to possess the quality and "5" being the highest degree.

    **I would like to have the following qualities:**

    | | | | | | |
    |---|---|---|---|---|---|
    | Persistence | 1 | 2 | 3 | 4 | 5 |
    | Honesty | 1 | 2 | 3 | 4 | 5 |
    | Responsibility | 1 | 2 | 3 | 4 | 5 |
    | Sense of humor | 1 | 2 | 3 | 4 | 5 |
    | Self-criticism | 1 | 2 | 3 | 4 | 5 |
    | Diligence | 1 | 2 | 3 | 4 | 5 |
    | Discipline | 1 | 2 | 3 | 4 | 5 |
    | Sociability | 1 | 2 | 3 | 4 | 5 |
    | Confidence | 1 | 2 | 3 | 4 | 5 |
    | Respectful of elders | 1 | 2 | 3 | 4 | 5 |
    | Organized | 1 | 2 | 3 | 4 | 5 |
    | Independence | 1 | 2 | 3 | 4 | 5 |
    | Inquisitiveness | 1 | 2 | 3 | 4 | 5 |
    | Original thinking | 1 | 2 | 3 | 4 | 5 |
    | Optimism | 1 | 2 | 3 | 4 | 5 |
    | Politeness | 1 | 2 | 3 | 4 | 5 |
    | Criticality | 1 | 2 | 3 | 4 | 5 |

**Thank you for your participation.**

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
