# Peer review of "Personality Development and Behavior in Adolescence: Characteristics and Dimensions"

_societies, doi:10.3390/soc13060148_

Round 1

Reviewer 1 Report

Dear authors,

Congratulations on the article! I think it is a well-written article that deserves to be published.

As a suggestion, in the first part of the article, in the ”Introduction” chapter, I think it would be good to motivate more strongly the importance of the theme and the place your work occupies in the specialized literature: this theme has been addressed before, with different methods or with the same methods, is the topic important for your society/country, etc.

Best regards

Author Response

Dear reviewer,

Thank you for the reviews sent, as well as the comments and
recommendations made in them to improve our paper. All revisions to the
manuscript  have been made using the "Track changes" function as required
by the journal

We have taken into account all the recommendations made in both reviews,
namely:

Introduction: the importance of the topic and the place of the article in
the research field, as well as its significance for society, are shown. The
objectives of the article are expanded, connections are made between the
topics covered to achieve a better understanding and overall impression.

Theoretical part: Made better transitions and connections between
individual parts.

The entire article has been grammar checked and proofread to improve
grammar and language.

Once again, thank you for your recommendation!

With Best Wishes,

The Authors

Reviewer 2 Report

First, I congratulate the authors for their efforts in carrying out the study. I would like to make some considerations and suggestions for improving the manuscript.

Title: I suggest modifying the title. The authors are not looking at child development, but rather issues related to personality development and behavior.

Summary: the summary should attract the reader's attention to the article. Therefore, the summary should be rethought. There is not enough data to understand the method and there is no description of results to figure out the conclusion.

Introduction: the first part of the introduction needs revision. There are sentences with different information that don't connect. The objective of the study appears at this moment but lacks textual coherence for a full understanding of the information and justifications of the authors. The reference of the authors used is missing.

In the part related to the theoretical basis, the authors described important issues for the basis of the study, but it requires textual cohesion. There are many pieces of information that need connection with each other.

Method: Is there ethical approval for carrying out the study? This information must be cited. What would be the characterization of the study?

The method should be further described. Are there inclusion and exclusion criteria for participants? How were they recruited?

There is a need to describe the tests used and also the research procedures, thinking that the method must be reproducible.

Discussion: Results indicated interesting aspects, however, the discussion only describes the findings, but does not relate them to the literature, or the hypotheses or considerations of the authors are not theoretically based, leaving the discussion very superficial. The authors found differences between the 3 populations that participated in the research and there is no justification, mainly because the Roma families obtained a great difference in relation to the others. The discussion should be reviewed and deepened to value the study and its findings.

Conclusion: I suggest adding research limitations and suggestions for future research.

I suggest reviewing aspects of grammatical cohesion, especially the introduction, and discussion.

Author Response

Dear reviewer,

Thank you for the reviews sent, as well as the comments and
recommendations made in them to improve our paper. All revisions to the
manuscript  have been made using the "Track changes" function as required
by the journal

We have taken into account all the recommendations made in both reviews,
namely:

Changed the title: “Personality development and behavior in adolescence:
characteristics and dimensions” to better match the content of the paper.

Abstract: It has been revised to make it more attractive and informative. A
description of the methods, results, sample, and conclusion is included.

Introduction: the importance of the topic and the place of the article in
the research field, as well as its significance for society, are shown. The
objectives of the article are expanded, connections are made between the
topics covered to achieve a better understanding and overall impression.

Theoretical part: Made better transitions and connections between
individual parts

Materials and methods: All aspects of ethical consent and approval from the
Ethics Committee and regulations were added, which were followed in order
to guarantee the ethical side of the information obtained and the methods
of conducting the field work.

The creation of the sample is presented in more details, as well as the
selection of respondents and the ways to ensure reliable and representative
information.

The tests and research procedures used are added.

Discussion: The obtained results are related to the literature. It is
justified, the confirmation of the hypotheses.

Conclusion: The research limitations and future author research that
furthers the topic under consideration are added.

The entire article has been grammar checked and proofread to improve
grammar and language.

Once again thank you for your time and all the recommendations!

With Best Wishes,

Tha authors

Round 2

Reviewer 2 Report

First, I would like to congratulate the authors' efforts to improve the manuscript.

I only have two points to highlight:

the abstract was very good, but the first part, which refers to the objectives of the manuscript and the study, could be unified.

at the beginning of the discussion, there is a phrase that is repeated: Respectful of elders, what would it be?

Author Response

Dear reviewers and editors,
Thank you for the reviews sent, as well as for the remarks and comments made in relation to improving the article.
We've made the following changes that were requested in the review:
- we unified the abstract and the introduction regarding the objectives of the study.
- we removed a phrase regarding "Respectful of elders" that is repeated in the discussion.
Thank you again
